# Exploring the Importance of Corticalization Occurring in Alveolar Bone Surrounding a Dental Implant

**DOI:** 10.3390/jcm11237189

**Published:** 2022-12-02

**Authors:** Marcin Kozakiewicz, Tomasz Wach

**Affiliations:** Department of Maxillofacial Surgery, Medical University of Lodz, 113 Żeromskiego Str., 90-549 Lodz, Poland

**Keywords:** dental implants, long-term results, long-term success, marginal bone loss, functional loading, intra-oral radiographs, radiomics, texture analysis, corticalization, bone remodeling

## Abstract

Several measures describing the transformation of trabecular bone to cortical bone on the basis of analysis of intraoral radiographs are known (including bone index or corticalization index, CI). At the same time, it has been noted that after functional loading of dental implants such transformations occur in the bone directly adjacent to the fixture. Intuitively, it seems that this is a process conducive to the long-term maintenance of dental implants and certainly necessary when immediate loading is applied. The authors examined the relationship of implant design features to marginal bone loss (MBL) and the intensity of corticalization over a 10-year period of functional loading. This study is a general description of the phenomenon of peri-implant bone corticalization and an attempt to interpret this phenomenon to achieve success of implant treatment in the long term. Corticalization significantly increased over the first 5-year functional loading (CI from 200 ± 146 initially to 282 ± 182, *p* < 0.001) and maintained a high level (CI = 261 ± 168) in the 10-year study relative to the reference bone (149 ± 178). MBL significantly increased throughout the follow-up period—5 years: 0.83 ± 1.26 mm (*p* < 0.001), 10 years: 1.48 ± 2.01 mm (*p* < 0.001). MBL and radiographic bone structure (CI) were evaluated in relation to intraosseous implant design features and prosthetic work performed. In the scope of the study, it can be concluded that the phenomenon of peri-implant jawbone corticalization seems an unfavorable condition for the future fate of bone-anchored implants, but it requires further research to fully explain the significance of this phenomenon.

## 1. Introduction

The use of dental implants is the primary method of replacing missing teeth. Nowadays, it is very widely modified [1,2,3] and applied from simple oral surgery [4] to very advanced craniomaxillofacial procedures [5,6,7,8]. This implant treatment has good long-term results, but still some implants are lost.

It has long been noted that bone apposition and remodeling processes occur around dental implants. Direct evidence of these phenomena is provided by dental implants removed after many years of their functional load [9]. Retrieval and histological analysis of dental implants for fracture or other reasons (such as orthodontic, psychological, esthetic, and hygienic reasons) [10] is able to explain the corticalization phenomenon induced by implants. Most of the present histological studies on human specimens find compact, lamellar bone with many Haversian systems and osteons near the implant surface with increased bone-implant contact (BIC) up to 60–90% in 7–8 years mean duration of functional loading [11,12,13,14,15,16,17,18,19,20,21,22,23,24,25,26,27,28,29,30,31]. It is also worth summarizing two well-known truths on the basis of these studies. First, loaded implants presented an average of 10% higher BIC when compared with unloaded ones. Second, approx. 10–12% higher BIC is reported for immediately loaded dental implants [32,33,34,35]. In loaded implants, transverse collagen fibers of the bone are more abundant, while in unloaded implants, these collagen fibers in bone tended to run in a more longitudinal way. Peri-implant bone is particularly thickened around the top of the threads [36]. Rougher surfaces have approx. a 10% higher area of bone apposition than machined surfaces. However, the Scandinavians, having introduced implants with a machined surface in the 1970s, still believe that these implants have superiority over implants with a moderately rough surface [37]. Multiple remodeled regions representing many remodeling cycles over the years are found in peri-implant bone. Ongoing apposition and resorption phenomena were present inside the threads. The osteocyte number is higher near loaded versus unloaded implants [10].

Moreover, it was already reported [38] that loading was able to stimulate bone remodeling at the interface, that a higher percentage of lamellar bone was found in loaded implants and more osteoblasts and osteoclasts were found in those loaded implants. The implant loading seemed to determine differences in the distribution of the bone collagen fibers too [14,39,40]. The transverse collagen fibers were mainly located at the lower flank of the threads, where compressive loads exerted their effects. Transverse collagen fibers have been described as the fibers most able to resist compressive loads, and this fact can explain their higher quantity in loaded than unloaded implants. A lower mineral density was present in the peri-implant bone around unloaded implants [41]. The loading forces direction could have determined a higher mineralization of the osseous tissue located in the coronal side of the threads when compared to that in an apical location [42].

The above-mentioned observations are probably so pronounced in the jaws because the bone appositional index here is one of the highest in the human body [43]. It is higher than in the iliac bone, femur or vertebrae. This process leads to the osseointegration of the dental implants firstly but later probably is responsible for the corticalization. The remodeling and the superimposition of new osteons on the older ones is found too [44].

Knowing that crestal bone is the basis for dental implants to function as intended [45,46], it seems that other factors such as gingival pocket, biotype, width of keratinized gingival zone, color and translucency of soft tissues [47,48,49] are secondary. In recent years, there seems to be a growing interest in the phenomenon of corticalization [50,51,52]. It has been hypothesized that almost no bone loss can be expected after bone remodeling over the implant neck [53]. It will be interesting to see whether the corticalization phenomenon affects the height of the bone supporting the implant. The question arises as to how this process is related to vertical bone loss and how it relates to the long-term success of implant treatment. For this, analysis of the microstructure of the bone surrounding the implant is needed [54], and the key place is the bone adjacent to the implant neck [55,56].

Fine bone morphology can be registered using microcomputed tomography as well as even using 3-tesla magnetic-resonance imaging (MRI) [57]. Magnetic resonance tomography instruments that trigger a field of 7 tesla have also been available for several years and are being used to analyze bone microstructure [58]. However, a series of limitations of these advanced technologies should be highlighted. In daily clinical practice, it is standard practice to use intraoral radiographs [59,60,61,62,63,64] or pantomographic radiographs [65,66,67,68] to analyze the condition of the peri-implant jawbone. Cone-beam tomographs [69] are used much less frequently. The cost of a 7-tesla scanner is not inconsiderable. There is little availability of this newly developed technology. There are no developed sequences for peri-implant jawbone imaging. Metal components such dental implants and parts of prosthetic work can create artifacts in MRI images and interfere with the diagnostic process, not to mention advanced studies of the bone structure at the implant wall [70], and most importantly, MRI is used to study the cancellous bone, not the structure of the cortical bone [71,72,73,74,75]. This is still a matter of the future [76], and for now, one can rely on imaging studies with the use of intraoral, periapical radiographs [77,78] for the reasons cited above.

The suspected long-term disadvantage of corticalization [79] is based on bone index (BI) analysis. There are some doubts about the specificity of this measure in detecting corticalization [80]. It seems that this measure does not discriminate very strongly between homogeneous dark areas (crestal bone loss) and homogeneous bright areas (corticalization of trabecular bone). Another inconvenience is the need to use the inverse of the bone index, i.e., 1/BI. Next, it is known that 1/BI is highest in bone loss regions, significantly lower in cortical bone and lowest significantly at the site of trabecular bone [80].

The aim of this study was to determine whether corticalization (basing on the corticalization index) in long-term follow-up is a negative phenomenon for the fate of dental implants.

## 2. Materials and Methods

The collected material is the result of prospective acquisition of radiological data during the clinical course of oral implantological treatment of patients with missing teeth in the maxillary and mandibular region. Inclusion criteria: at least 18 years of age, bleeding on gingival probing < 20%, probing depth ≤ 3 mm, good oral hygiene, regular follow-ups, following doctor’s orders. Exclusion criteria: uncontrolled internal co-morbidity (diabetes mellitus, thyroid dishormonoses, rheumatoid disease and other immunodeficiencies), a history of oral radiation therapy, past or current use of cytostatic drugs, soft tissue augmentation, low quality or lack of follow-up radiographs. General health was confirmed via anamnesis and evaluation of body mass index (BMI) using a serum test of thyrotropin, calcium and triglycerides (the way to describe their general condition, i.e., emanation of the health status on entry to the study). Finally, clinical and radiological data of 911 persons were included in this study.

The dental implants were inserted by one dentist (M.K.) according to the protocols recommended by the manufacturers. A total of 22 types of dental implant were used in this study: AB Dental Devices I5 (www.ab-dent.com (accessed on 21 July 2022), Ashdod, Israel) 102 pieces, ADIN Dental Implants Touareg (www.adin-implants.com (accessed on 21 July 2022), Afula, Israel) 89 pieces, Alpha Bio ARRP (www.alpha-bio.net (accessed on 21 July 2022), Petah-Tikva, Israel) 14 pieces, Alpha Bio ATI (www.alpha-bio.net (accessed on 21 July 2022), Petah-Tikva, Israel) 139 pieces, Alpha Bio DFI (www.alpha-bio.netv (accessed on 21 July 2022), Petah-Tikva, Israel) 43 pieces, Alpha Bio OCI (www.alpha-bio.net (accessed on 21 July 2022), Petah-Tikva, Israel) 28 pieces, Alpha Bio SFB (www.alpha-bio.net (accessed on 21 July 2022), Petah-Tikva, Israel) 62 pieces, Alpha Bio SPI (www.alpha-bio.net (accessed on 21 July 2022), Petah-Tikva, Israel) 448 pieces, Argon K3pro Rapid (www.argon-dental.de (accessed on 21 July 2022), Bingen am Rhein, Germany) 182 pieces, Bego Semados RI (www.bego-implantology.com (accessed on 21 July 2022), Bremen, Germany) 12 pieces, Dentium Super Line (www.dentium.com (accessed on 21 July 2022), Gyeonggi-do, South Korea) 38 pieces, Friadent Ankylos C/X (www.dentsplysirona.com (accessed on 21 July 2022), Warszawa, Poland) 14 pieces, Implant Direct InterActive (www.implantdirect.com (accessed on 21 July 2022), Thousand Oaks, United States of America) 139 pieces, Implant Direct Legacy 3 (www.implantdirect.com (accessed on 21 July 2022), Thousand Oaks, United States of America) 48 pieces, MIS BioCom M4 (www.mis-implants.com (accessed on 21 July 2022), Bar-Lev Industrial Park, Israel) 8 pieces, MIS C1 (www.mis-implants.com (accessed on 21 July 2022), Bar-Lev Industrial Park, Israel) 307 pieces, MIS Seven (www.mis-implants.com (accessed on 21 July 2022), Bar-Lev Industrial Park, Israel) 921 pieces, MIS UNO One Piece (www.mis-implants.com (accessed on 21 July 2022), Bar-Lev Industrial Park, Israel) 40 pieces, Osstem Implant Company GS III (www.en.osstem.com (accessed on 21 July 2022), Seoul, South Korea) 15 pieces, SGS Dental P7N (www.sgs-dental.com (accessed on 21 July 2022), Schaan, Liechtenstein) 12 pieces, TBR Implanté (www.tbr.dental (accessed on 21 July 2022), Toulouse, France) 6 pieces, and Wolf Dental Conical Screw-Type (www.wolf-dental.com (accessed on 21 July 2022), Osnabrück, Germany) 31 pieces. The total number of introduced dental implants was 2700 pieces. The appearance of the tested implants is shown in Figure 1.

All implants were loaded late, i.e., min. 3 months after the implants were placed in the bone. Standardized intraoral radiographs [81] were taken immediately before prosthetic restoration (initial radiograph), 5 and 10 years later. Focus X-ray apparatus (Instrumental Dental, Tuusula, Finland) was set to the constant technical parameters: exposure time 0.1 s, voltage in the lamp 70 kV and current 7 mA. An intraoral parallel technique was used. To ensure an identical relative position of the implant, an X-ray tube and radiation detector and a set of RINN XCP rings and holders were utilized (Dentsply International Inc., Cheung Sha Wan, Hong Kong, China) with a silicone bite index. The video part of the system was a recording plate with a photosensitive storage surface (Digora Optime digital radiography system—Soredex, Tuusula, Finland [61]). Immediately after the X-ray exposure, the storage phosphor plate was placed in a scanner that reads radiographic information (the image size was 476 × 620 pixels; the pixel size was 70 μm × 70 μm). A computer coupled with the scanner processed, presented and archived acquired images. Patients included in the study were followed by a single dentist during the entire period. The average marginal bone loss (MBL) of the alveolar crest after osseointegration (initial) at 5 and 10 years of functional loading was measured. In addition, the bone texture features at these time periods were calculated. The influence of factors related to implant design (Table 1) was evaluated.

In the radiographs obtained in this way, a region of interest (ROI) was established in the area of bone near the implant neck (Figure 2, green). The second ROI was established in an image of intact bone distant from the dental implant (it was referent bone, yellow). The surface area of each ROI was 1500 pixels squared.

Radiologically recorded peri-implant bone structure was studied via digital texture analysis using the corticalization index previously proposed [80] as version 1 (CI). It consists of the product of a measure that evaluates the number of long series of pixels of similar optical density with the mean optical density of the studied site (in the numerator) and the magnitude of the chaotic arrangement of the texture pattern, i.e., differential entropy (in the denominator).

The texture of X-ray images was analyzed in MaZda 4.6 freeware invented by the University of Technology in Lodz [82] to test measures of corticalization in the per-implant environment of trabecular bone (representing original bone before implant-dependent alterations) and soft tissue (representing product of marginal bone loss). MaZda provides both first-order (mean optical density) and second-order (differential entropy: DifEntr, long-run emphasis moment: LngREmph) data. Due to the fact that the second-order data are given for four directions in the image and in the present study the authors do not wish to search for directional features, the arithmetic mean of these four primary data was included for further analysis. The regions of interest (ROIs) were normalized (*μ* ± 3*σ*) to share the same mean (*μ*) and standard deviation (*σ*) of optical density within the ROI. To eliminate noise [83] further, worked on data were reduced to 6 bits. For analysis in a co-occurrences matrix, a spacing of 5 pixels was chosen. In the formulas that follow, *p*(*i*) is a normalized histogram vector (i.e., histogram whose entries are divided by the total number of pixels in ROI), *i* = 1,2,…, *N_g_*, and *N_g_* denotes the number of optical density levels. The mean optical density feature (only a first order feature) was calculated as below:(1)$$\mathrm{Mean}\;\mathrm{Optical}\;\mathrm{Density}=\sum_{i=1}^{N_{g}}ip\left(i\right)$$

Second order features:(2)$$\mathrm{DifEntr}=-\sum_{i=1}^{Ng}p_{x-y}\left(i\right)log\left(p_{x-y}\left(i\right)\right)$$
where Σ is the sum, *N_g_* is the number of levels of optical density in the radiograph, *i* and *j* are the optical density of pixels 5 pixels distant one from another, *p* is the probability and *log* is the common logarithm [54]. The differential entropy calculated in this way is a measure of the overall scatter of bone structure elements in a radiograph. Its high values are typical for cancellous bone [64,84,85,86]. Next, the last primary texture feature was calculated:(3)$$\mathrm{LngREmph}=\frac{\sum_{i=1}^{Ng}\sum_{k=1}^{Nr}k^{2}p\left(i,k\right)}{\sum_{i=1}^{Ng}\sum_{k=1}^{Nr}p\left(i,k\right)}$$
where Σ is the sum, *N_r_* is the number of series of pixels with density level *i* and length *k*, *N_g_* is the number of levels for image optical density, *N_r_* is the number of pixel in the series and *p* is the probability [87,88]. This texture feature describes thick, uniformly dense, radio-opaque bone structures in intra-oral radiograph images [84,86].

The equations for mean optical density, DifEntr and LngREmph were subsequently used for the corticalization index (CI) construction [80]:(4)$$\mathrm{LngREmph}=\frac{\sum_{i=1}^{Ng}\sum_{k=1}^{Nr}k^{2}p\left(i,k\right)}{\sum_{i=1}^{Ng}\sum_{k=1}^{Nr}p\left(i,k\right)}$$

Statistical analysis includes feature distribution evaluation, mean (*t*-test) or median (W-test) comparison, analysis of regression and one-way analysis of variance or the Kruskal-Wallis test as no-normal distribution or between-group variance indicated significant differences in investigated groups. Detected differences or relationships were assumed to be statistically significant when *p* < 0.05. Statgraphics Centurion version 18.1.12 (StatPoint Technologies, Warrenton, VA, USA) was used for statistical analyses.

## 3. Results

In the implantological material collected, it was found at baseline (initial) that 86.7% of the implants were not affected by marginal bone loss at all and 13.3% of the implants had some degree of bone loss (in this subgroup the MBL was 1.93 ± 1.85 mm). After 5 years, bone loss was not present in 54.4% of the implants, and bone loss was noted in 43.6% of the implants (in this subgroup the MBL was 1.91 ± 1.26 mm). At the final point of the study, i.e., after 10 years of functional loading, there was zero bone loss in 44.4% of implants, while 55.6% were affected to some degree by marginal bone loss (in this subgroup the MBL was 2.67 ± 2.04 mm).

A sequential, significant increase in the CI in peri-implant bone was observed from the initial study (i.e., just after functional loading) to five years (*p* < 0.001). Subsequently, a slight decrease in the CI was noted at the ten-year study (*p* < 0.05), but the CI is significantly higher than on the day functional loading began. When analyzing MBL, it was found to progress statistically significantly throughout the study with high significance (*p* < 0.001). When examining the relationship between CI and MBL, it was noticed that the two variables were associated with each other from five years after the functional loading of the dental implants, i.e., at the fifth year: CC = 0.11, R^2^ = 1.2%, *p* < 0.001, and at the tenth year: CC = 0.12, R^2^ = 1.4%, *p* < 0.01. MBL was directly proportionally related with an increase in the CI (Table 2 and Figure 3).

It is important to evaluate corticalization in relation to basic epidemiological data. Hence, the relationship with gender, smoking, location, etc. is shown below (Table 3).

Sex, pre-prosthetic surgical augmentation procedures and surgical technique for augmentation are not a differentiating factor for the study population at any stage of the survey. In contrast, the opposite is true of localization. For smokers with implants put in the mandible or posterior part of the dental arch, corticalization is higher than in the smoker group (excluding 5-year observation) with implants in the maxilla or posterior dental arch, throughout the study period. The association of increasing weight, height and BMI (as well as serum calcium levels) in patients with a decreasing corticalization index can be seen. On the other hand, increasing age (but no relation found in 10-year investigation) and thyrotropin levels in the patients studied are accompanied by an increasing corticalization index (Figure 4).

The results obtained for the different types of implants that remained under long-term follow-up are shown below (Figure 5). They are arranged in all four graphs from the implant type with the lowest peri-implant bone corticalization to the highest corticalization (for both CI and MBL). It is noticeable that MBL does not correspond directly to CI values for individual implants. Therefore, further analyses were performed in groups organized differently, i.e., according to the features of the implant designs (Table 4) and the prosthetic restoration used (Table 5).

In the group of implants made of grade 5 titanium alloy, lower corticalization was noted in the initial period, which increased significantly at 5 years. A higher MBL occurred in the later observation periods. Throughout the observation period, implants inserted subcrestally have the lowest MBL. However, surprisingly, during the initial period, the lowest bone loss is accompanied by the highest level of corticalization. These differences disappear at the five-year follow-up period, and then at the ten-year follow-up period the relationships reverse—the lowest corticalization is with subcrestally inserted implants as also the MBL is the lowest. One-piece implants (i.e., of the “Custom” connection type contrary to internal connection) were characterized by higher MBL up to and including the fifth year of observation. This is not followed by the CI value. Evaluating the connection shape, it could be seen that corticalization is greatest with one-piece implants (i.e., without a socket for the abutment), but this does not go hand in hand with the MBL value at five and ten years. When the implants do not have a haed microthread, higher MBL values are recorded with them. CI values are also elevated, but statistically insignificant. The shape of the implant body has no effect on corticalization and marginal loss at either the initial or the 10-year follow-up period. It was only noted that in the fifth year of functional loading there was less corticalization and more bone loss with tapered implants. The lowest MBL and highest CI were noted in implants with a V shape and reverse butteress thread. The lowest bone loss supported by an increased CI occurred with flat apex implants. Only increasing bone loss over time was observed in implants with apex groove. This was not followed by increasing CI values.

Platform switching induces less corticalization of peri-implant bone, as do single-crown restorations. Such restorations have low MBLs, but the lowest MBLs are found in cases of bridges. Implant-supported bridges initially have a high CI, but at 10-year follow-up, corticalization is already lower than in splinted crowns and overdentures.

## 4. Discussion

The healing process and osseointegration in dental implants is a dynamic phenomenon. When an implant is installed, the next surgical procedure causes some marginal bone loss [89]. Within the initial healing phase, the recruitment and migration of osteoprogenitor cells to the surface of the implant occurs. During the secondary healing phase, new bone is apposited. Next, the peri-implant bone is reabsorbed and replaced with a new viable bone, i.e., remodeling is featured [32,33,34,35,44,89]. In cases of successful treatment, this reaction reaches a balance with the patient’s body, and only in disequilibrium does the MBL increase, thereby damaging peri-implant bone [90]. Pinpointing what underlies this dysfunction is crucial for current dental implantology.

In a long-term study [91], assessment of corticalization in peri-implant bone was performed only visually (Figure 2 in Buser’s study) and described in the 10-year data as “well-corticalized”. In the current state of development of image analysis methods, a much more precise description can be obtained [82,92]. However, this is a high-quality study, and the authors collected results depicting the corticalization phenomenon. This can be seen in the radiological figures, e.g., Figure 9 in Ref. [91], where MBL is preceded in the bone by a pronounced disappearance of trabeculation and an increase in bone density. However, the authors did not point out the corticalization. Similarly, this is seen in Figure 5 in Albrektsson’s publication [93]. Today, the phenomenon of bone density increase can be analyzed qualitatively and, of course, in relation to the MBL [44,45]. Similar interesting illustrative material can be found in another 10-year follow-up study [37] where there are clear features of severe peri-implant bone corticalization in their Figure 1b. Unfortunately, the corticalization term is not used at all by the authors. This is probably due to the purpose of the paper and the lack of publications analyzing bone texture at dental implants in detail. One can also find a publication based on radiographic material, which does not include a single X-ray in the text [94]. In this case, it is impossible to determine what the authors faced in their study. The second issue is the use of simple quantitative measures (they do not describe the internal state of the bone), e.g., the percentage of implant surface remaining in contact with the jawbone (bone-to-implant contact, BIC) or the amount of marginal bone loss from the alveolar crest (MBL). Intuitively, it seems that bone quality (structure testing) is important in the long-term maintenance of dental implants [95].

Corticalization (and associated marginal bone loss) related with the type of implant used is not easy to interpret but is definitely the result of the aforementioned balance and bone remodeling. It probably depends on the type of implant, but implant selection is not random. It depends on the bone conditions and the possibility of using prosthetic solutions in a given implantological system, which correspond to a given dento-gnathic status. Finally, certainly, it depends on the dentist’s preferences for using a particular implant system. The results presented here are derived from these many influencing factors, but this is a typical situation in everyday clinical work and hence worth considering and trying to understand.

It is now known from everyday clinical work that implant treatment is very long term or even over a lifetime [96]. It seems that the changes in peri-implant bone structure observed at this time are not a simple projection of the occlusal load in the bone [40], but a complex modulation of osteoimmunological activity [97,98,99]. Recently, it has been noticed that mechanotransduction may promote the alteration of bone marrow monocyte activation. Thus, occlusal force may modulate the osteoimmunity in peri-implant bone [100]. In addition, there is a synergy between mechanical loading and the signaling pathway for macrophage function, which is related to the αM integrin controlling the activity of the mechanosensitive ion channel Piezo1 [101] and the genetically determined bone reaction [102]. Further confirmation of an osteoimmodulatory mechanism, rather than a simple loading reaction, in peri-implant bone remodeling is the positive role of topically applied bisphosphonates in reducing MBL [103,104,105]. In the near future, a biological analysis approach combining genomic with clinical data including bone structure will be able to explain the mechanism of corticalization [102].

The arrangement of implant types from causing the least peri-implant bone corticalization at 5 and 10 years to the implant causing the most corticalization does not reflect the same arrangement of implant types relative to marginal bone loss (Figure 5). Thus, the relationship is not a simple one of the type given implant = defined bone loss, and yet, this would be supported by the corticalization index value. However, when considering all 2700 implants, the association of corticalization with marginal bone loss is statistically highly significant (*p* < 0.001). Therefore, the study material here was divided differently (see Table 4). The names of the implants were discarded, and the design features were taken, and thus, the implants were combined into groups with common design features.

It is interesting to note that one-piece implants are not associated with the smallest MBLs, despite not having a micro-gap or the possibility of bacterial contamination in the gingival sulcus and junctional epithelium [106]. Perhaps this is due to the fact that these implants are narrower than two-part implants and can be used in a narrower alveolar crest. Probably, the smaller volume of the bone base is prone to atrophy due to limited bone vascularization and mechanical reasons even though there is no contamination from the microgap. On the other hand, it is not surprising that implants inserted subcrestally have low MBL and low corticalization values [107]. Considering the 10-year follow-up period, these 2 characteristics indicate a good prognosis for subcrestal implants. Prosthetic work placed on such implants leaves adequate biological space for good marginal periodontal function [108], and there is certainly more bone around them from the start than if one-piece implants are used. This ensures permanent maintenance of the peri-implant bone level [109]. When considering the significance of the micro-thread implant neck, it should be noted that the MBL observed in this study is slightly lower than in studies known from the literature [110,111]. At the same time, these studies here confirmed the effectiveness of micro-thread use in minimizing MBL over a 10-year period of functional loading. However, there was no significant change in the peri-implant jawbone cortication of micro-thread implants. The interaction of thread parameters has a significant influence on the peak compressive and tensile strains at the cancellous as well cortical bone. Body-related parameters are more effective on the peak compressive strain at the cortical interface only [112]. The results of this work here seem to confirm these results from the numerical analysis. CI and MBL proceed independently of the implant body, or in other words, alternative further features determine corticalization and marginal bone loss (general health, osteopenia, sarcopenia, dietary supplements taken, drug or behavioral weight loss, details of prosthetic work, occlusion, parafunctions, history of prosthetic repairs, additional dental treatment, saliva composition and active protein content, overactive tongue, etc.). The high MBL (and disparate CI results) observed with rounded apex hole implants seems to be more related to the fact that they are cylindrical implants without threads and with no modifications in the neck area rather than to the effect of the apex hole on the condition of the neck peri-implant bone.

Single crowns do not cause bone structure changes around the implants on which they are set. At the same time, they characterize low marginal bone loss. In cases loaded with bridges, lower measures of corticalization and lower MBL were noted at 5 years than in overdentures and compared to splinted crowns at 10 years. In the case of works using switching platforms, it was noted that corticalization values are always lower than in works without prosthetic platform switching. No differences were noted in terms of MBL. Among the multitude of implant design features and series of prosthetic solutions considered, it should be noted that the lowest long-term bone loss was observed in cases of implant loading with bridges. In contrast, the highest MBL was recorded in cases of splinted crowns. These changes were accompanied by corresponding CI values (higher in high MBL and lower in low MBL). Surprisingly, platform switching was not noted to affect MBL, but there was a significantly lower CI with such implants. However, MBL in the platform-switched prosthetic was lower than total MBL at the 10-year follow-up.

Marginal bone loss has been postulated to have a multi-factorial etiology [113] and can be considered to occur early or late in the lifetime of an implant. It is certain that within the first year after placement, MBL observed is a consequence of bone remodeling subsequent to surgical and prosthetic work [56] as well early loading challenges undertaken by an implant and its associated prosthesis [113,114]. It has been known for a long time that smoking as well as previous history of periodontitis are associated with peri-implantitis and may represent risk factors for this disease [115]. Given the role of adaptive bone remodeling, corticalization may be influenced by infection as a barrier for oral microflora invasion. Over the longer term, the cumulative effect of chronic etiological factors that are immunological, environmental, patient-related factors such as motivation, smoking, para- or disfunctions, infection and inflammation, as well the influence of the surgeon or prosthodontist can affect the increase of corticalization and bone loss in long-term observation [113,114,116,117]. Due to the poorly studied phenomenon of corticalization in dental implantology, the authors speculate that the phenomenon of increased bone structure density itself may be heterogeneous. They would not be surprised if it turns out that some specific form of corticalization or the degree of its severity may be prognostically favorable, while another form may be unfavorable, as appears to be the case after this study.

Corticalization index increasing with age (although observed in 5-year follow-up), rising TSH levels and decreasing serum calcium levels seem to support the negative significance of the peri-implant bone corticalization phenomenon. These selected markers similarly behave with the progression of the aging process [118,119]. In this regard, the matter of interpreting the observed phenomenon in bone is not clear. Only a narrow fragment of possible systemic effects on bone has been examined. However, it is a contribution to further interesting research.

The conclusions of this work cannot be radical. The suspicion of corticalization as an unfavorable predictor for the development of marginal bone loss is based on several hundred implants observed over a 10-year period. This, unfortunately, confirms previous suspicions [79]. Undoubtedly, to establish more certain relationships [120,121,122], studies should be conducted on a larger number of implants. This kind of research is prompted by the relationship noted here between low MBL and surprisingly high CI with V shape and reverse butteress threaded implants. Multicenter studies are also needed. Different surgical and prosthetic protocols are worth testing. Both authors of this study believe that other (easier and widely available) techniques for assessing the corticalization phenomenon in peri-implant bone should also be tried.

## 5. Conclusions

In the scope of the study, it can be concluded that the phenomenon of peri-implant jawbone corticalization clearly seems to be a condition that is unfavorable for the future fate of bone-anchored implants.

## Figures and Tables

**Figure 1 jcm-11-07189-f001:**
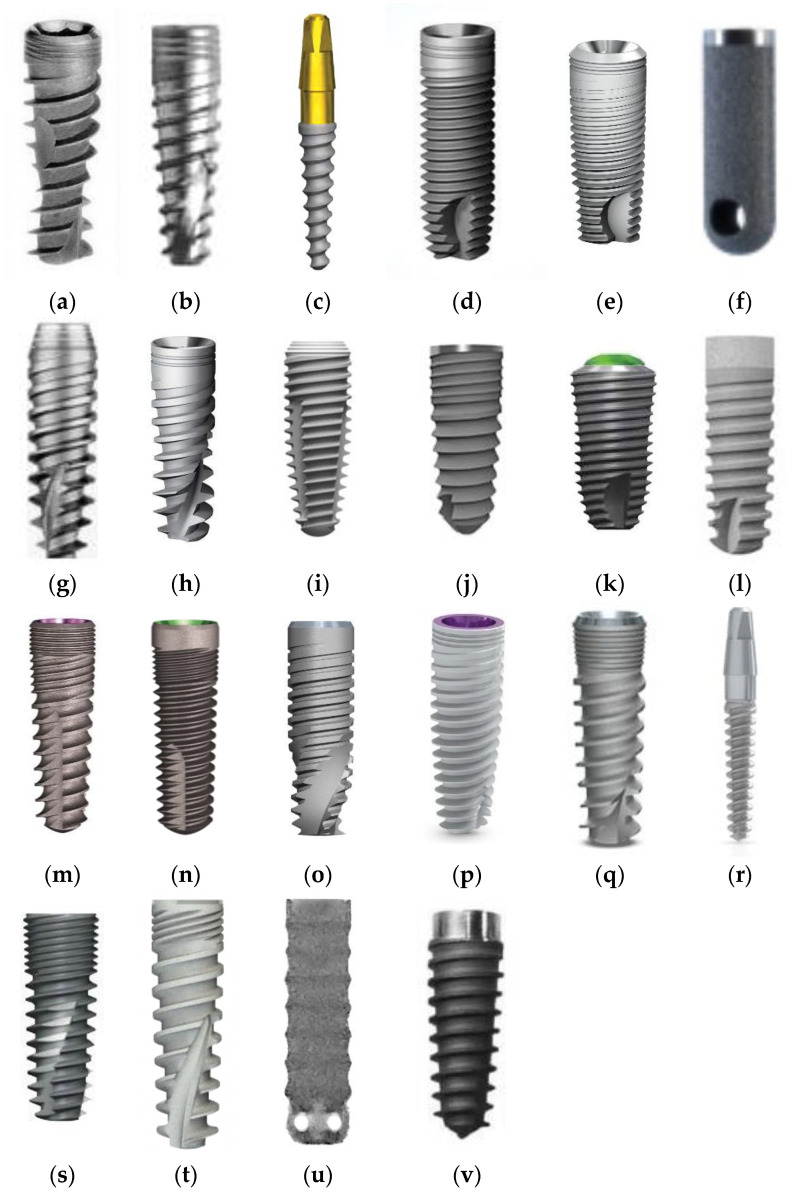
The appearance of the dental implants compared in this study, in alphabetical order: (**a**) AB Dental Devices I5; (**b**) ADIN Dental Implants Touareg; (**c**) Alpha Bio ARRP; (**d**) Alpha Bio ATI; (**e**) Alpha Bio DFI; (**f**) Alpha Bio OCI; (**g**) Alpha Bio SFB; (**h**) Alpha Bio SPI; (**i**) Argon Medical Productions K3pro Rapid; (**j**) Bego Semados RI; (**k**) Dentium Super Line; (**l**) Friadent Ankylos C/X; (**m**) Implant Direct InterActive; (**n**) Implant Direct Legacy 3; (**o**) MIS BioCom M4; (**p**) MIS C1; (**q**) MIS Seven; (**r**) MIS UNO One Piece; (**s**) Osstem Implant Company GS III; (**t**) SGS Dental P7N; (**u**) TBR Implanté; (**v**) Wolf Dental Conical Screw-Type.

**Figure 2 jcm-11-07189-f002:**
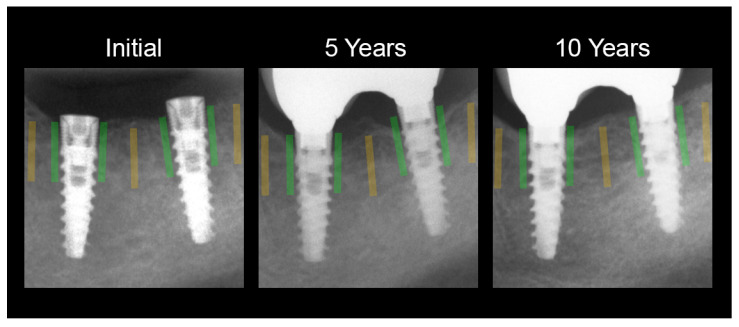
Image data acquisition method for texture analysis in intraoral radiographs. ROIs highlighted in yellow are sites in the alveolar crest distant from the dental implants (reference). ROIs marked in green are sites examined along the neck portion of the implants and represent, respectively: radiographs taken immediately prior to prosthetic work—initial ROI; radiographs taken after five years of functional loading—5 years ROI; radiographs taken after ten years of functional loading—10 years ROI. The data extracted from these ROIs were later analyzed in freeware MaZda 4.6 [79,80,82] and used to calculate the corticalization index.

**Figure 3 jcm-11-07189-f003:**
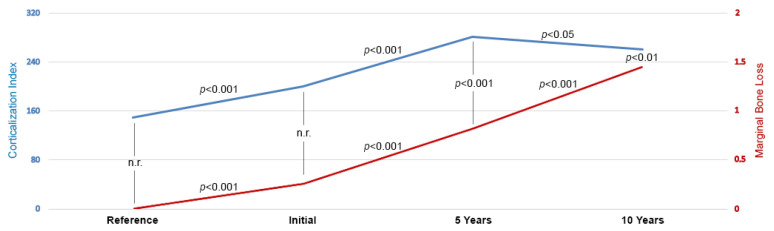
The results of peri-implant bone corticalization assessment (corticalization index, blue line, data without a unit) and marginal bone loss (red line, data in millimeters). There was a statistically significant increase in the values of both variables at each stage of the study. Moreover, it was noted that there was a directly proportional relationship of marginal bone loss with the progression of corticalization at 5 years and 10 years of functional loading of the implants. Abbreviations: n.r.—no relationship.

**Figure 4 jcm-11-07189-f004:**
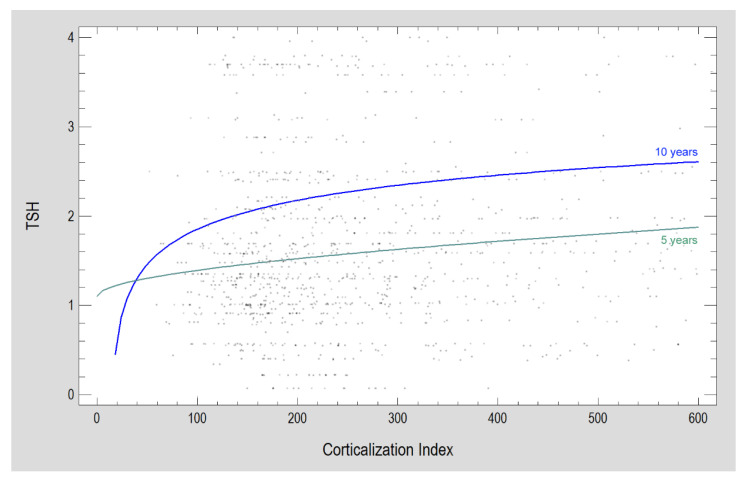
An example of the relationship found between patients’ general condition (TSH: thyrotropin serum level in mU/L) and the corticalization index. Both relationships are statistically significant (*p* < 0.05).

**Figure 5 jcm-11-07189-f005:**
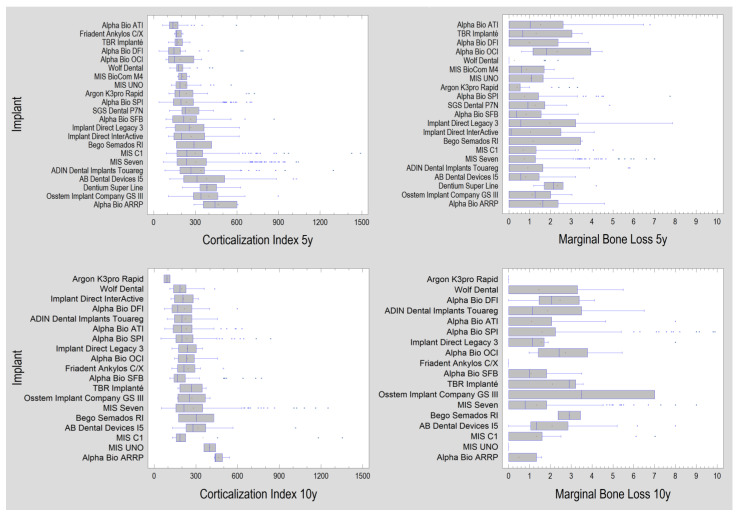
The results obtained for the types of dental implants studied. The charts on the left show the results of the corticalization evaluation (five and ten years). The results here are arranged from lowest mean (**top**) to highest mean (**bottom**). On the right are the results of marginal bone loss arranged in the same order as for corticalization—it can be seen that bone loss does not absolutely correspond to corticalization.

**Table 1 jcm-11-07189-t001:** Design features of dental implant used in this study (www.spotimplant.com/en/ (access on 21 July 2022)). Alphabetical order of the implant names.

Manufacturer Implant Type	Titanium Alloy	Level	Connection Type	Connection Shape	Neck Shape	Neck Microthreads	Body Shape	Body Threads	Apex Shape	Apex Hole	Apex Groove
AB Dental Devices I5	Grade 5	Bone Level	Internal	Hexagon	Straight	No	Tapered	Square	Flat	No Hole	Yes
ADIN Dental Implants Touareg	Grade 5	Bone Level	Internal	Hexagon	Straight	Yes	Tapered	Square	Flat	No Hole	Yes
Alpha Bio ARRP	Grade 5	Tissue Level	Custom	One Piece Abutment	Straight	No	Tapered	Reverse Buttress	Cone	No Hole	No
Alpha Bio ATI	Grade 5	Bone Level	Internal	Hexagon	Straight	Yes	Straight	Square	Flat	No Hole	Yes
Alpha Bio DFI	Grade 5	Bone Level	Internal	Hexagon	Straight	Yes	Tapered	Square	Flat	No Hole	Yes
Alpha Bio OCI	Grade 5	Bone Level	Internal	Hexagon	Straight	No	Straight	No Threads	Dome	Round	No
Alpha Bio SFB	Grade 5	Bone Level	Internal	Hexagon	Straight	No	Tapered	V Shaped	Flat	No Hole	Yes
Alpha Bio SPI	Grade 5	Bone Level	Internal	Hexagon	Straight	Yes	Tapered	Square	Flat	No Hole	Yes
Argon Medical Prod. K3pro Rapid	Grade 4	Subcrestal	Internal	Conical	Straight	Yes	Tapered	V Shaped	Dome	No Hole	Yes
Bego Semados RI	Grade 4	Bone Level	Internal	Hexagon	Straight	Yes	Tapered	Reverse Buttress	Cone	No Hole	Yes
Dentium Super Line	Grade 5	Bone Level	Internal	Conical	Straight	No	Tapered	Buttress	Dome	No Hole	Yes
Friadent Ankylos C/X	Grade 4	Subcrestal	Internal	Conical	Straight	No	Tapered	V Shaped	Dome	No Hole	Yes
Implant Direct InterActive	Grade 5	Bone Level	Internal	Conical	Straight	Yes	Tapered	Reverse Buttress	Dome	No Hole	Yes
Implant Direct Legacy 3	Grade 5	Bone Level	Internal	Hexagon	Straight	Yes	Tapered	Reverse Buttress	Dome	No Hole	Yes
MIS BioCom M4	Grade 5	Bone Level	Internal	Hexagon	Straight	No	Straight	V Shaped	Flat	No Hole	Yes
MIS C1	Grade 5	Bone Level	Internal	Conical	Straight	Yes	Tapered	Reverse Buttress	Dome	No Hole	Yes
MIS Seven	Grade 5	Bone Level	Internal	Hexagon	Straight	Yes	Tapered	Reverse Buttress	Dome	No Hole	Yes
MIS UNO One Piece	Grade 5	Tissue Level	Custom	One Piece Abutment	Straight	No	Tapered	Square	Dome	No Hole	Yes
Osstem Implant Company GS III	Grade 5	Bone Level	Internal	Conical	Straight	Yes	Tapered	V Shaped	Dome	No Hole	Yes
SGS Dental P7N	Grade 5	Bone Level	Internal	Hexagon	Straight	Yes	Tapered	V Shaped	Flat	No Hole	Yes
TBR Implanté	Grade 5	Bone Level	Internal	Octagon	Straight	No	Straight	No Threads	Flat	Round	Yes
Wolf Dental Conical Screw-Type	Grade 4	Bone Level	Internal	Hexagon	Straight	No	Tapered	V Shaped	Cone	No Hole	Yes

**Table 2 jcm-11-07189-t002:** The progressive increase in the difference in bone structure of implant-loaded versus reference cancellous bone and the observed relationship with marginal bone loss.

Region of Interest/Period	Corticalization Index	Marginal Bone Loss [mm]	Simple Regression
Reference Cancellous Bone	149 ± 178	0.00 ± 0.00	n.a.
Initial Peri-Implant Observation	200 ± 146	0.25 ± 0.94	n.s.
5 Years Peri-Implant Observation	282 ± 182	0.83 ± 1.26	CC = 0.11; R^2^ = 1.2%; *p* < 0.001
10 Years Peri-Implant Observation	261 ± 168	1.48 ± 2.01	CC = 0.12; R^2^ = 1,4%; *p* < 0.01

Abbreviations: n.a.—not applicable; n.s.—no statistical significance; CC—correlation coefficient: R^2^—coefficient of determination.

**Table 3 jcm-11-07189-t003:** Presentation of included population. Assessment of the impact of baseline epidemiological data on the corticalization index observed in peri-implant bone.

Clinical Feature	Option/Value of the Feature	Corticalization Index		
		Initial	5 Years	10 Years
Sex	Female	205 ± 169	279 ± 176	263 ± 151
	Male	194 ± 114	285 ± 190	260 ± 190
Tobacco Smoking	Non-Smoker	200 ± 152 ^L^	283 ± 185	257 ± 166 ^L^
	Smoker	203 ± 91 ^H^	272 ±155	301 ± 184 ^H^
Jaw	Maxilla	175 ± 108 ^L^	239 ± 151 ^L^	223 ± 148 ^L^
	Mandible	190 ± 179 ^H^	336 ± 203 ^H^	302 ± 179 ^H^
Localization in Dental Arch	Anterior	166 ± 92 ^L^	247 ± 163 ^L^	226 ± 162 ^L^
	Posterior	212 ± 174 ^H^	295 ± 188 ^H^	273 ± 169 ^H^
Jawbone Status	Augmented	220 ± 210	267 ± 164	263 ± 142
	Intact	193 ± 116	286 ± 188	261 ± 176
Augmentation Technique	Implant Neck Bone Chips	236 ± 269	292 ± 187	271 ± 133
	Implant Neck Bone Substitute	183 ± 107	210 ± 138	280 ± 211
	Bone Substitute Sinus Lift	210 ± 143	248 ± 138	252 ± 135
Age	47 ± 13 years	Direct Relation *	Direct Relation *	No Relation
Patient Height	1.70 ± 0.09 m	No Relation	No Relation	Inverse Relation *
Patient Weight	75 ± 19 Kg	No Relation	Inverse Relation *	Inverse Relation *
Body Mass Index	26 ± 4	No Relation	Inverse Relation *	Inverse Relation *
Serum Thyrotropin	1.73 ± 1.07 mU/L	Direct Relation *	Direct Relation *	Direct Relation *
Total Serum Calcium	2.39 ± 0.61 mmol/dL	Inverse Relation *	Inverse Relation *	Inverse Relation *
Serum Triglycerides	1.24 ± 0.57 mmol/L	Direct Relation *	No Relation	No Relation

^H^ value higher than in other implant design options within observation period (*p* < 0.05); ^L^ value lower than in other implant design options within observation period (*p* < 0.05); * means significant relationship (*p* < 0.05) between corticalization index and the clinical quantitative (i.e., numerical) feature.

**Table 4 jcm-11-07189-t004:** Peri-implant bone feature observed among examined implant designs groups.

Design Parameter	Option	Feature	Initial	5 Years	10 Years
Titanium Alloy *n* = 2196	Grade 4	MBL	0.00 ^L^	0.00 ^L^	0.00
		CI	184 ^H^	179 ^L^	189
	Grade 5	MBL	0.00 ^H^	0.00 ^H^	0.91
		CI	163 ^L^	225 ^H^	209
Immersion Level *n* = 2196	Subcrestal	MBL	0.00 ^L^	0.00 ^L^	0.00 ^L^
		CI	198 ^H^	181	201 ^L^
	Bone Level	MBL	0.00	0.00	0.97 ^H^
		CI	163 ^L^	224	205 ^L^
	Tissue Level	MBL	0.00 ^H^	1.24 ^H^	0.00
		CI	154	222	439 ^H^
Connection Type *n* = 2196	Internal	MBL	0.00 ^L^	0.00 ^L^	0.91
		CI	167	221	205 ^L^
	Custom	MBL	0.00 ^H^	1.24 ^H^	0.00
		CI	154	222	439 ^H^
Connection Shape *n* = 2196	Conical	MBL	0.00	0.00	0.00
		CI	202 ^H^	225	200 ^L^
	Internal Hexagon	MBL	0.00	0.00	0.97
		CI	151 ^L^	220	205 ^L^
	Internal Octagon	MBL	0.00	0.67	2.91
		CI	205	168	268
	One Piece Abutm	MBL	0.00	1.24	0.00
		CI	154	222	439 ^H^
Head Microthreads *n* = 2196	Yes	MBL	0.00	0.00 ^L^	0.73 ^L^
		CI	170 ^H^	221	201
	No	MBL	0.00	0.61 ^H^	1.15 ^H^
		CI	158 ^L^	222	227
Body Shape *n* = 2196	Tapered	MBL	0.00	0.00 ^L^	0.85
		CI	167	226 ^H^	206
	Straight	MBL	0.00	1.33 ^H^	1.15
		CI	172	147 ^L^	206
Body Threads *n* = 1760	Butteress	MBL	0.00	2.15 ^H^	n.a.
		CI	190	383 ^H^	n.a
	Reverse Butteress	MBL	0.00 ^L^	0.00 ^L^	0.79 ^L^
		CI	171 ^H^	239 ^H^	213
	V Shape	MBL	0.00 ^L^	0.00 ^L^	0.00 ^L^
		CI	174 ^H^	197 ^L^	184
	Square	MBL	0.00 ^L^	0.00 ^L^	0.91 ^L^
		CI	150 ^L^	201 ^L^	211
	No Threads	MBL	0.30 ^H^	1.54 ^H^	2.57 ^H^
		CI	190	164 ^L^	232
Apex Shape *n* = 2196	Cone	MBL	0.00	0.00	0.00
		CI	122 ^L^	199	193
	Dome	MBL	0.00	0.00	0.79
		CI	174 ^H^	230 ^H^	213
	Flat	MBL	0.00	0.45 ^H^	1.21
		CI	148	103 ^L^	201
Apex Hole *n* = 1447	Round	MBL	0.00 ^L^	1.54 ^H^	2.57 ^H^
		CI	190	164 ^L^	232
	No or other	MBL	0.30 ^H^	0.00 ^L^	0.79 ^L^
		CI	167	221 ^H^	206
Apex Groove *n* = 2196	Yes	MBL	0.00 ^L^	0.00 ^L^	0.79 ^L^
		CI	167	220	105
	No	MBL	0.00 ^H^	1.66 ^H^	2.00 ^H^
		CI	154	297	258

^H^ value higher than in other implant design options within observation period (*p* < 0.05); ^L^ value lower than in other implant design options within observation period (*p* < 0.05); *n*—number of evaluated dental implants; MBL—marginal bone loss is given as median due to non-normal distribution in mm; CI—corticalization index is given as median due to non-normal distribution.

**Table 5 jcm-11-07189-t005:** The prosthetic works examined in this study.

Prosthetic	*n*	Feature	Initial	5 Years	10 Years
Single Crown	734	MBL	0.00 ^H^	0.00	0.91
		CI	153 ^L^	196 ^L^	186 ^L^
Splinted Crowns	794	MBL	0.00	0.00	1.20 ^H^
		CI	198	224	227 ^H^
Bridge	576	MBL	0.00 ^L^	0.00 ^L^	0.00 ^L^
		CI	172 ^H^	251 ^H^	215
Overdenture	160	MBL	0.00	0.49 ^H^	0.00 ^L^
		CI	185 ^H^	392 ^H^	239 ^H^
Platform Switching	509	MBL	0.00	0.00	1.06
		CI	155 ^L^	197 ^L^	200

^H^ value higher than in other prosthetic solutions (*p* < 0.05); ^L^ value lower than in other prosthetic solutions (*p* < 0.05); *n*—number of evaluated dental implants; MBL—marginal bone loss is given as median due to non-normal distribution in mm; CI—corticalization index is given as median due to non-normal distribution.

## Data Availability

The data on which this study is based will be made available upon request at https://www.researchgate.net/profile/Marcin-Kozakiewicz (accessed on 22 October 2022).

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
