# Peer review of "Exploring the Importance of Corticalization Occurring in Alveolar Bone Surrounding a Dental Implant"

_jcm, 2022, doi:10.3390/jcm11237189_

Round 1
Reviewer 1 Report
This is a great manuscript, but it needs a modeling approach such as finite element method. That could be make it more general technique for further and more extensive future studies. The clarification of the method is not so satisfying which I would suggest a little more detail and more extensive discussion.
Author Response
Please, find attached response [in bleu text]. Author

Reviewer 2 Report
1. Page 1, Line 37-39: “It has been hypothesized that almost no bone loss can be expected after bone remodeling over the implant neck.” What about cases with peri-implantitis? Bone loss can extend beyond implant platform.
2. What was the reason of reviewing MRI in the introduction section when it was not actually used in the present study? Moreover, what is the difference with a 3-tesla versus 7-tesla MRI? Not all dental implant field readers are familiar with MRI, please elaborate.
3. Please provide more background knowledge of the alveolar bone corticalization and peri-implant health.
4. The present data and study would be more valuable to include clinical measurements such as probing depth and bleeding on probing or suppuration. Clinical measurements are also very important to peri-implant conditions, what was the reason not being able to include clinical measurements in the present manuscript?
a. Were all the implants included in the study healthy? Were there any peri-implantitis or peri-implant mucositis implants included?
5. Figure 1: the initial and the 10 years peri-apical films showed more radiopacity on the implant fixtures compared to the 5 years x-ray. Were the exposure settings similar in all three x-rays?
6. Figure 1: the 5 years follow up green ROI included areas above the crestal alveolar bone. Would this affect the analysis?
7. How was the marginal bone loss measured? Please kindly elaborate in the material and method section. Was there any customized x-ray mount used during the exposure of the radiographs? The marginal bone loss amount reported in the results are mostly <2mm, and with a little shift of the x-ray angle can influence the data collection greatly. For instance, the X-ray angle between the three films in Figure 1 are all slightly different. How did the authors overcome this problem when analyzing the xrays?
8. When was the blood test (Serum Thyrotropin, Total Serum Calcium, Serum Triglycerides) data collected? Was the data collected at initial visit or later? Please also include this information in the material and methods section.
Author Response

(The authors gave the same response as above.)

Reviewer 3 Report
Thanks for the extensive study of various implant systems over a follow up period of a decade. There is no doubt that implant macro and micro surfaces have effect on osseointegration and various levels of marginal bone loss, after certain period of implants loading.
It would be interesting to consider the surface characteristic (chemical/molecular, hydrophilicity and hydrophobic conditions etc.) rather focusing on only implant surface design.
There is a need to address the surface characteristics of various implant systems used in the study.
Author Response

(The authors gave the same response as above.)
